# Automatic classification of medical image modality and anatomical location using convolutional neural network

**Chen-Hua Chiang** [1,2], **Chi-Lun Weng**[3], **Hung-Wen Chiu**[4]*

**1** Department of Radiology, Shuang Ho Hospital, Taipei Medical University, New Taipei City, Taiwan,
**2** Department of Radiology, School of Medicine, College of Medicine, Taipei Medical University, Taipei,
Taiwan, **3** Tainan Municipal Hospital, Tainan, Taiwan, **4** Graduate Institute of Biomedical Informatics,
College of Medical Science and Technology, Taipei Medical University, Taipei, Taiwan

* hwchiu@tmu.edu.tw

Automatic classification of medical image modality
and anatomical location using convolutional neural
network. PLoS ONE 16(6): e0253205. https://doi.
org/10.1371/journal.pone.0253205

Technology: VIT University, INDIA

**Data Availability Statement:** All relevant data are
within the manuscript.

**Funding:** The author(s) received no specific
funding for this work.

## Abstract

Modern radiologic images comply with DICOM (digital imaging and communications in medicine) standard, which, upon conversion to other image format, would lose its image detail
and information such as patient demographics or type of image modality that DICOM format
carries. As there is a growing interest in using large amount of image data for research purpose and acquisition of large amount of medical image is now a standard practice in the clinical setting, efficient handling and storage of large amount of image data is important in both
the clinical and research setting. In this study, four classes of images were created, namely,
CT (computed tomography) of abdomen, CT of brain, MRI (magnetic resonance imaging) of
brain and MRI of spine. After converting these images into JPEG (Joint Photographic
Experts Group) format, our proposed CNN architecture could automatically classify these 4
groups of medical images by both their image modality and anatomic location. We achieved
excellent overall classification accuracy in both validation and test sets (> 99.5%), specificity
and F1 score (> 99%) in each category of this dataset which contained both diseased and
normal images. Our study has shown that using CNN for medical image classification is a
promising methodology and could work on non-DICOM images, which could potentially
save image processing time and storage space.

## Introduction

Modern radiologic medical images, comply with DICOM (digital imaging and communications in medicine) standard, which is a worldwide standard for the storage and transmission
of medical imaging. The reason for using the DICOM standard is to ensure that all the medical
images made by different machines, hospitals or companies can speak the same language and
therefore operate within the same environment. Each DICOM file has a header containing
data such as patient demographic information, acquisition parameters and image dimensions.
The remaining portion of the DICOM file contains the image data [1]. The header information
is encoded within the DICOM file so that it cannot be accidentally separated from the image

**Competing interests:** The authors have declared that no competing interests exist.

data. If the header is separated from the image data, the image will not be displayed properly [2]. However, there are times that these DICOM images would be converted into different format such as JPEG (Joint Photographic Experts Group) or PNG (Portable Networks Graphics) for reasons such as they are of smaller file size or supporting website display. In such scenario, the information that the DICOM image carries would be lost and unable to retrieve. Among these other file format, JPEG is the most popular format as it can be read by all computer platforms. However, as JPEG format uses lossy compression, it may suffer from compression artifacts or loss of details which may affect image interpretation, and is therefore generally avoided for diagnostic purpose.

With the advancement of medical imaging technique, acquisition of large, high-resolution images has now become a standard expectation and requirement in diagnostic setting by both clinician and radiologist. As there is a continuous interest on medical images research using machine learning methods, a need to pre-process and classify large volume of image data is growing since assigning human to recognize, sort and organize every medical image is laborious and error-prone [3]. What is more, efficient handling of large amount of image data is also important in the setting of teleradiology [4]. Although we could use text-based retrieval method to categorize medical images since DICOM header contains information on body part and image modality, previous research have shown that about 15% of images contained false tag entries which can result in wrong categorization [5]. Therefore, a reliable pure image-based classification algorithm would be applicable in both clinical and research setting.

Deep learning (DL) is a breakthrough technique that can extract discriminative image features. Convolutional Neural Network (CNN) is the most commonly used DL architecture in many image analyses and computer vision-related tasks [6]. DL can automatically learn visual features from the input image pixels [7] through mechanisms of combining multiple (deep) layers of receptive fields, and pooling [8]. It has been shown to have a superior performance compared to other conventional machine learning methods such as SVM and ANN [9, 10]. The goal of this study is to see if CNN can accurately discriminate medical images from their imaging modalities and anatomic location after they have been converted into other image format, namely JPEG format. JPEG format was chosen as it is the most portable image format with small file size and can substantially reduce storage requirements and expedite image transfer and training for machine learning purpose but is known for its inadequate image quality for diagnosis.

## Materials and methods

### Dataset

The institutional review board approved this retrospective study (Taipei Medical University-Joint Institutional Review Board No: N201912079), with a waiver of informed consent. We focused on CT and MRI images as these two image modalities are the main workhorse in diagnostic imaging. For each image modality, two anatomical locations were selected and thus a total of four medical image sub-groups were collected and they were CT of abdomen, CT of brain, MRI of brain and MRI of lumbar spine. Examples of these four categories of images are shown (Fig 1).

We first searched in the PACS system of Shuang-Ho Hospital for these four types of medical images and then randomly select the images from different patients and saved them into JPEG files (Fig 2). JPEG format was chosen for the reason stated previously and also, many of the publicly accessible medical image dataset are of JPEG format. We saved all image slices as separate JPEG images. As our database contains both normal and pathological images, it should therefore be generalized enough to represent what radiologists or researchers face in

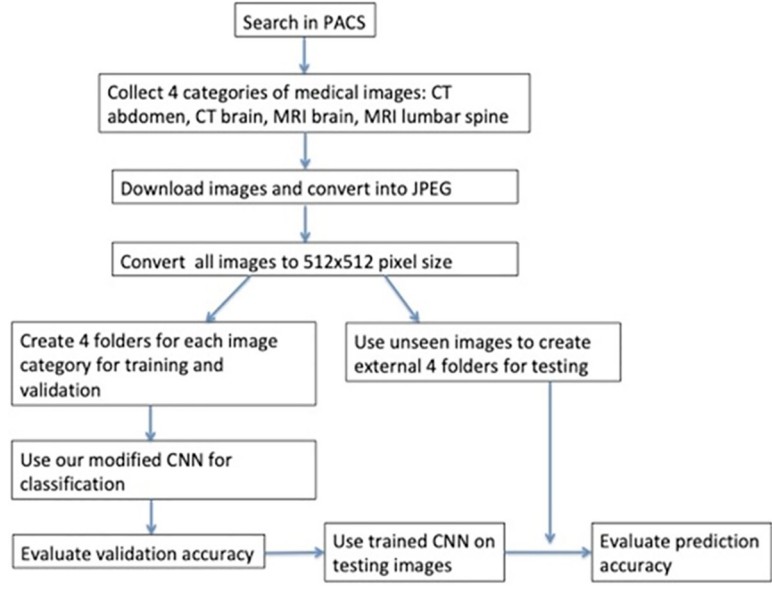

**Fig 1. Example image of each modality and anatomical location.** (a) abdomen CT, (b) brain CT, (C) brain MRI, (d) MRI lumbar spine.

daily practice. MRI was performed with various machines, including 1.5T (Signa HDX 1.5T, GE Healthcare, Milwaukee, WI, USA) and 3T scanners (Discovery MR750 3T, GE Healthcare, Milwaukee, WI, USA). The CT was performed with two machines in our hospital, Bright Speed 16 CT scanner (GE Healthcare, Milwaukee, WI, USA) and Discovery CT 750 HD scanner (GE Healthcare, Milwaukee, WI, USA).

For the MRI of the lumbar spine group, T1, T2 sequences with both sagittal and axial views were collected. For the MRI of the brain group, images of T1, T2 axial views, T2 coronal view and DWI images were collected. For the abdominal CT group, only axial views were selected but both the images of with and without contrast enhancement were collected. For the brain CT group, only the non-enhanced brain CT images and axial view were collected. All images were in grayscale. After converting these medical images from DICOM format into JPEG images, we resized them into 512x512 in pixels. Default window level of the images was downloaded from our PACS and no pre-processing or normalization of these images was performed.

A dataset containing abovementioned four images folders were created with folder names as CT of abdomen, CT of brain, MRI of brain and MRI of lumbar spine for the CNN to read. The amount of images of each folder was as below: total of 742 images for CT of abdomen folder, 739 images for the CT of brain folder, 707 images for the MRI of brain folder and 690 images for the MRI of spine folder (Table 1).

**Fig 2. Workflow chart.**

Table 1. Data distribution across the image modalities.

| Category | Total Samples | Training | Validation | Test | Pixel size | File Type |
|---|---|---|---|---|---|---|
| Abdominal CT | 742 | 476 | 223 | 50 | 512x512 | JPEG |
| Brain CT | 739 | 517 | 222 | 50 | 512x512 | JPEG |
| Lumbar Spine MRI | 690 | 483 | 207 | 50 | 512x512 | JPEG |
| Brain MRI | 707 | 495 | 212 | 50 | 512x512 | JPEG |

## CNN architecture

Our proposed model was modified from a CNN architecture provided by Matlab 2020a (The MathWorks, Inc., Natick, MA) for hand-written digit data classification. The original CNN architecture was modified to read image size of 512-by-512-by-1, with 512 being the pixel size of images and 1 indicating grayscale image. The data was randomly divided into training and validation sets using 70% of the images for training and 30% for validation. The training set was used to train the algorithm, and the validation set was for tuning of hyperparameters. After multiple attempts, further modification was made for the CNN to contain five sets of convolutional layers followed by batch normalization layer, ReLu layer and then Max Pooling layer. The filter size for the all five convolutional layers were 3-by-3 but we gradually increased the number of filters so that there were 8 filters in the first convolutional layer, 16 filters in the second, 24 filter in the third, 32 filters in the fourth and 48 filters in the fifth convolutional layers. Default padding of one and stride of one in the convolutional layer were used. The fully connected layer was set to four as there were four classes to be classified (Fig 3).

The trained network then predicted the labels of the validation data and calculated accuracy, which is the fraction of labels that the network predicts correctly. The network used stochastic gradient descent with momentum (SGDM) with an initial learning rate of 0.01. The maximum number of epochs was adjusted to 8 after several attempts to gain best result.

To further test the accuracy of our trained network, we created a test set containing these four classes of images, with each class containing 50 new and unseen images (Fig 2) to our algorithm. We calculated the prediction accuracy of these unseen data using the same method as we did with the validation data. The above experiments were conducted with a standard MacBook Pro computer incorporated with Intel Core i7 2.7 GHz CPU, 16G main memory and Intel HD Graphics 530 1536 MB.

## Statistical analysis

A confusion matrix was created from our predicted result and compared with actual results to calculate the following evaluation metrics:

$$\text{Accuracy} = (TP + TN) \div \text{Total} \tag{1}$$

$$\text{Precision} = TP \div (TP + FP) \tag{2}$$

$$\text{Recall} = \text{Sensitivity} = TP \div (TP + FN) \tag{3}$$

$$\text{Specificity} = TN \div (FP + TN) \tag{4}$$

$$\text{F1 Score} = 2 \times \text{Precision} \times \text{Recall} \div (\text{Precision} + \text{Recall}) \tag{5}$$

where TP is the true positive count, FP is the false positive count, FN is the false negative count and FP is the false positive count.

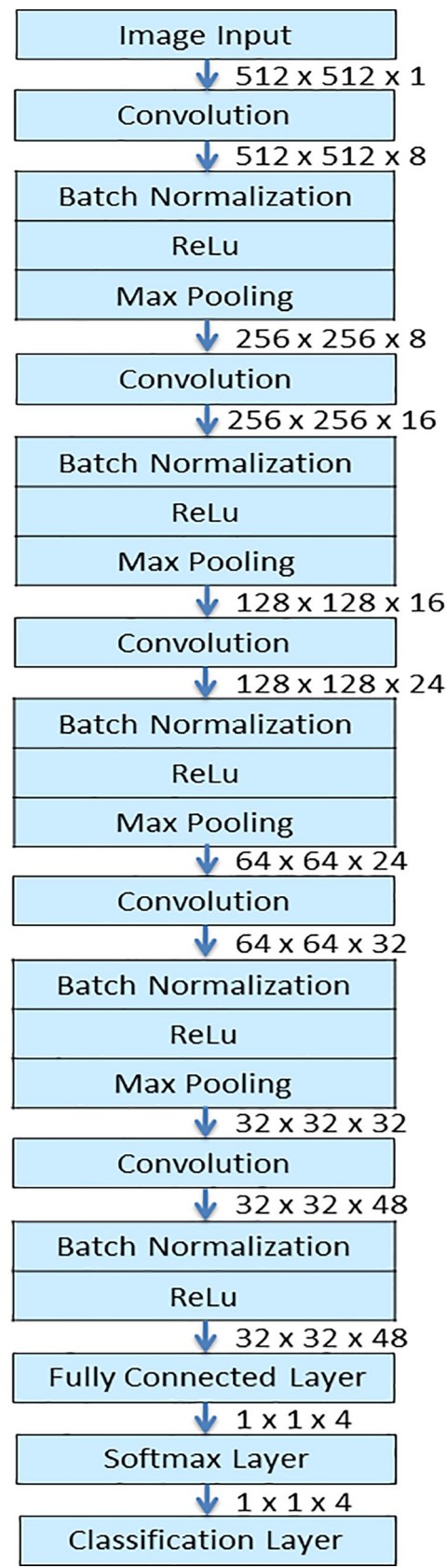

**Fig 3. Our CNN architecture.**

**Table 2. Performance of our CNN architecture on image modality and anatomical location classification.**

| | Number of Attempts | | | | | |
|---|---|---|---|---|---|---|
| | 1st | 2nd | 3rd | 4th | 5th | Average |
| Elapsed Time (m = min, s = second) | 23m25s | 22m53s | 23m4s | 22m1s | 21m51s | 22m39s |
| Validation Accuracy (%) | 99.77 | 100 | 99.88 | 99.88 | 100 | 99.91 |
| Testing Accuracy (%) | 100 | 99.5 | 100 | 98.5 | 100 | 99.6 |

Accuracy is the ratio of the correctly labeled images to the whole pool of images. Precision measures how precise or accurate our model is, that is, how many images of our model's prediction matched with actual ground truth label. Recall is also named as sensitivity which is the ratio between how many images that were correctly identified as one class of image to how many images that were actually that class of image. Specificity is the ratio between how many images that were correctly classified as not belonging to a class of image to the amount of total images that were actually not that class of images. F1 score considers both precision and recall and is the harmonic mean (average) of the precision and recall. F1 score suits best if there is some sort of unbalance between precision and recall in the system and is considered a better indicator of the classifier's performance than the measure of accuracy.

## Results

We trained our model and repeated this task for 5 times, for each run, the time elapsed for the training and validation took about 21 to 24 minutes. The average validation accuracy of these 5 attempts was 99.91%. We then used our trained network for prediction on the testing set of unseen images, after 5 runs, the model achieved an average accuracy rate of 99.6% (Table 2).

During validation, among the four groups of medical images, there were cases of CT of brain image being misclassified as MRI of brain image, cases of MRI of brain image misclassified as MRI of spine image and cases of MRI of spine image misclassified as CT of brain image (Table 3). On the testing set, only the images of spine MRI were misclassified as MRI of brain images (Table 4).

The results of accuracy, precision, sensitivity, specificity and F1 score for these four classes in both the validation and testing sets were presented and showed that the accuracy, precision, recall, specificity and F1 score were all near 100% in the validation set (Table 5). The result dropped slightly in the testing set which showed a precision of 98.4% in the MRI brain group and recall of 98.4% in the MRI of spine group (Table 6). The F1 score of both sets therefore were 99% and 99.2% respectively. All the other performance measurements were well above 99%. Examples of misclassified images by the validation and testing sets are shown in Fig 4.

## Discussion

In this digital age, many medical images may be converted into different file format to be carried around and for various purposes such as education or consultation. Converting DICOM

**Table 3. Multi-class confusion table for validation set of the five attempts.**

| | | Actual Class | | | |
|---|---|---|---|---|---|
| | | CT Abdomen | CT Brain | MRI Brain | MRI Spine |
| Predicted Class | CT Abdomen | 223 ± 0 | 0 | 0 | 0 |
| | CT Brain | 0 | 221.6 ± 0.24 | 0 | 0.2 ± 0.16 |
| | MRI Brain | 0 | 0.4 ± 0.24 | 211.8 ± 0.16 | 0 |
| | MRI Spine | 0 | 0 | 0.2 ± 0.16 | 206.8 ± 0.16 |

**Table 4. Multi-class confusion table for testing set of the five attempts.**

| | | Actual Class | | | |
|---|---|---|---|---|---|
| | | **CT Abdomen** | **CT Brain** | **MRI Brain** | **MRI Spine** |
| Predicted Class | CT Abdomen | $50 \pm 0$ | 0 | 0 | 0 |
| | CT Brain | 0 | $50 \pm 0$ | 0 | 0 |
| | MRI Brain | 0 | 0 | $50 \pm 0$ | $0.8 \pm 1.36$ |
| | MRI Spine | 0 | 0 | 0 | $49.2 \pm 1.36$ |

images into other file format also has its necessity in both clinical and research setting as it allows for faster image transmission and reduces file storage space. However, after conversion, they would lose their DICOM header information and one can no longer identify the image location and modality without expert's help, which is the first step in using these images for various purposes. Using human expert's visual inspection for image classification can be laborious and error-prone. Our result showed that using our simple convolutional neural network, we could achieve very high performance in classifying both image modality and anatomical location on JPEG images. This can be useful in the clinical setting such as teleradiology as one could use JPEG for its advantage on faster image transmission and restore them to remove the compression artifact associated with JPEG images as proposed by Chung et al [4]. Also, earlier work done by Szot et al have shown that using both JPEG- and JPEG2000-compressed images from the digital camera and a view-box allow readings of sufficient quality for the diagnosis of tuberculosis on chest X-ray, this means that using JPEG image for diagnostic purpose could suffice under specific condition and appropriate setting.

A reliable, automatic, pure image-based classification algorithm also has its demand in the research setting. Pizarro et al developed a deep learning algorithm using CNN architecture to automatically infer the contrast of MRI scans based on the image intensity of multiple slices for database management purpose in their research center. The MRI contrast dataset at their second part of study included fluid-attenuated inversion recovery (FLAIR), proton-density weighted, T1-weighted, post-contrast agent, T1-weighted pre-contrast agent, T2-weighted, high resolution T1-weighted, magnetic transfer ON, and magnetic transfer OFF. The CNN algorithm they developed could automatically identify the MRI contrast of previously described dataset with a less than 0.2% error rate, which was markedly lower than the random forest algorithm that had 1.74% error rate for the same MRI dataset [11]. A similar idea was investigated upon by Remedios et al for a similar purpose, in which they developed a novel 3D deep CNN-based method for MR image contrast classification. Specifically, they investigated on three classification tasks. Firstly, to identify T1-weighted, T2-weighted, and FLAIR contrasts. The second task was to identify pre versus post-contrast T1-weighted images. The third task was to identify pre versus post-contrast FLAIR images. They achieved a mean accuracy of 97.57% across the 3 tasks [3]. Compared with our work, their work focused on classifying different image contrast or sequences of the same image modality, namely MRI, and the same anatomical location, namely brain. Remedios et al showed some examples of the misclassified

**Table 5. Overall performance measures of our CNN architecture on validation set (In percentage).**

| | Accuracy | Precision | Recall | Specificity | F1 Score |
|---|---|---|---|---|---|
| CT Abdomen | 100 | 100 | 100 | 100 | 100 |
| CT Brain | 99.9 | 99.9 | 99.8 | 100 | 100 |
| MRI Brain | 99 | 99.8 | 99.9 | 99.9 | 100 |
| MRI Spine | 100 | 99.9 | 99.9 | 100 | 99.9 |

**Table 6. Overall performance measures of our CNN architecture on testing set (In percentage).**

|  | Accuracy | Precision | Recall | Specificity | F1 Score |
|---|---|---|---|---|---|
| CT Abdomen | 100 | 100 | 100 | 100 | 100 |
| CT Brain | 100 | 100 | 100 | 100 | 100 |
| MRI Brain | 99.6 | 98.4 | 100 | 99.5 | 99 |
| MRI Spine | 99.6 | 100 | 98.4 | 100 | 99.2 |

images by their proposed algorithm, and they were due to either severe artifact or pathologies. This is similar in our study in which one of our misclassified images had intracerebral hemorrhage.

For the task of image classification, CNN has been reported to outperform other conventional machine learning methods by Maruyama et al. They reported using CNN, namely AlexNet, Support Vector Machine (SVM) and Artificial Neural Network (ANN) to classify 3 different image modalities, namely, X-ray, CT and MRI images and amongst two image formats, namely, DICOM and JPEG format. They showed that CNN had 100% classification accuracy using the JPEG dataset and the accuracy rate was 94.4% for SVM, and 88.9% for ANN. The classification accuracy, using the DICOM dataset, was 100% for CNN, SVM, and ANN. Moreover, they reported a shorter processing time for one JPEG image (7 seconds) compared with DICOM image (10 seconds) for CNN [10]. This study could further justify the advantage of choosing CNN as the algorithm for pure image-based classification and JPEG as the image format.

There are several other studies that focused on classifying medical images according to the anatomic location or organ-specific anatomy. For example, Ning-ning Ren et al used image processing method to automatically recognize radiographic position of brain, cervical spine, chest, lumbar spine, pelvis, and limbs. The radiographic position was determined from frequency similarity and amplitude classification. The body region recognition was performed by image matching in the whole-body phantom image which they built a series of templates to compare with. Using these methods, they achieved an average organ recognition accuracy of 93.78% and the average judgment time was 0.29 s [12].

Hiroyuki Sugimori focused on classifying CT images according to their anatomic location by using various numbers of images slices ranging from 100 to 10000. They used AlexNet and GoogLeNet to classify 10 classes, namely, brain, neck, chest, abdomen, and pelvis with contrast-enhanced and plain images and found that the best overall accuracy was 0.721 for the classification of 10 classes by using GoogLeNet and 10000 image slices. Also, the best overall accuracy for the classification of the slice position without contrast media was 0.862 by using 2000 image slices dataset with AlexNet [13].

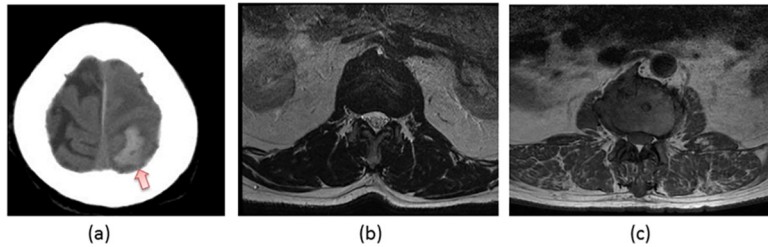

**Fig 4. Example of incorrect prediction by our proposed CNN architecture is shown.** (a) CT of brain image misclassified as MRI of brain image in the validation set. (b) MRI of spine image misclassified to CT of brain in the validation set. (c) MRI of spine image misclassified to MRI of brain in testing set. Notice the image in (a) contains intracerebral hemorrhage (arrow).

**Table 7. Summary of medical image classification on image modality and/or anatomy by various authors.**

| Author | Topic | Method Used | Main Evaluation Metrics |
|---|---|---|---|
| Pizarro et al | MRI Contrast Classification | CNN | Error Rate |
| Remedio et al | MRI Contrast Classification | 3D CNN | Accuracy Rate |
| Maruyama et al | Image Modality Classification | CNN, SVM, ANN | Accuracy Rate |
| Ren et al | Identify Radiographic position | Image Processing Method | Accuracy Rate |
| Sugimori | Classify Anatomic Location on CT Image | CNN | Accuracy Rate |
| Khan et al | Classify both Anatomic Organ and Image Modality | CNN | Accuracy Rate |
| Roth et al | Classify Anatomy on CT Image | CNN | Error Rate |

Sameer Khan et al tried to identify the same organs (lung, liver, heart, kidney and lumbar spine) from five different imaging modalities (CT, MRI, Positron emission tomography, ultrasound and X-ray images) by using a modified CNN architecture based on AlexNet. They achieved a validation and test accuracy of 76.6% and 81% respectively. This result outperformed three milestone CNN architectures namely, AlexNet, LeNet and GoogleLeNet which had test accuracy of 74%, 59% and 45% respectively [14]. Their result showed that the architectures used for natural image classification cannot be generalized on medical images and modification of the basic CNN architecture could yield better results for the task of medical image anatomy classification. This result is similar to our findings that modification of hyperparameters was required for medical image classification after we adopted the CNN architecture that was originally designed for hand-written digit data. Holger R. Roth et al developed a convolutional network with five layers of convolution filters to identify 5 classes of anatomy in CT images namely, neck, lungs, liver, pelvis and legs. Their testing set error rate improved from 9.6% prior to data augmentation to 5.9% after data augmentation [15, 16] Compared to their work, our study had a much higher accuracy rate (> 99.5%) but we focused on identifying anatomic location rather than specific anatomic organs.

Despite all previous works, to our knowledge, there were no previous studies that used CNN to classify medical images by both image modality and anatomic location, which is most practical in both clinical and research setting. The higher than 99.5% accuracy rate and the higher than 99% F1 score and specificity for each category of our validation and testing set demonstrated that using CNN for medical image classification is a promising application in real life scenario. A comprehensive table of previous works regarding medical image classification on either anatomy and/or image modality is summarized in Table 7.

There were several limitations in our study. Firstly, we only used JPEG images for our CNN architecture. It would have been interesting to compare the accuracy rate between JPEG and DICOM images using the same algorithm as the conversion of DICOM to JPEG format resulted in information loss which may impact the classification accuracy rate. However, Maruyama et al have already shown that there was no difference in classification accuracy between JPEG and DICOM images using CNN and that using JPEG image benefited from a shorter processing time. Therefore, we would assume that the difference in accuracy would have been small if this comparison was made. Secondly, we did not perform any image pre-processing for this study and only used the preset window level for our model. However, our excellent classification accuracy result could justify for the lack of image pre-processing and proved that using the preset window level was enough to perform this task well which also saved image processing time. Thirdly, we did not choose all the sequences in the two MRI classes, only those we thought was representative on that certain image modality and anatomic location, which might hinder its accuracy rate if applying this CNN architecture to other sequences in real life scenario. Fourthly, a limited number of test set examples were included in this study.

Fifthly, only 4 categories out of two image modalities and three anatomical locations were selected in this study, this could hinder the ability of generalization in the real life.

## Conclusion

In this study, we proposed a deep CNN architecture for the classification of medical image by both their anatomic location and image modality using JPEG images and achieved excellent overall classification accuracy in both validation and test sets ($> 99.5\%$). We also achieved very high specificity and F1 score ($> 99\%$) in each category of our dataset which contained both diseased and normal images. Our study confirmed that using CNN for medical image classification is a promising methodology and could work on non-DICOM images. This could potentially save image processing time and storage space in real life scenario. Future work on other anatomic locations and imaging modalities should be investigated upon in the hope that a fully automated medical image classification algorithm can be used in both clinical and research setting.

## Author Contributions

**Conceptualization:** Chen-Hua Chiang, Hung-Wen Chiu.

**Data curation:** Chen-Hua Chiang, Chi-Lun Weng.

**Formal analysis:** Chen-Hua Chiang.

**Investigation:** Chen-Hua Chiang, Chi-Lun Weng, Hung-Wen Chiu.

**Methodology:** Chen-Hua Chiang, Chi-Lun Weng.

**Project administration:** Chi-Lun Weng.

**Supervision:** Hung-Wen Chiu.

**Writing – original draft:** Chen-Hua Chiang.

**Writing – review & editing:** Hung-Wen Chiu.

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
