## [Decision Letter · Decision Letter 0]

4 May 2021

PONE-D-21-11867

Automatic Classification of Medical Image Modality and Anatomical Location Using Convolutional Neural Network

PLOS ONE

Dear Dr. Chiu,

Thank you for submitting your manuscript to PLOS ONE. After careful consideration, we feel that it has merit but does not fully meet PLOS ONE’s publication criteria as it currently stands. Therefore, we invite you to submit a revised version of the manuscript that addresses the points raised during the review process.

Based on the comments received from the reviewers and my own observation, I suggest major revisions for the paper.

We look forward to receiving your revised manuscript.

Kind regards,

Thippa Reddy Gadekallu

Academic Editor

PLOS ONE

Journal Requirements:

Reviewers' comments:

Reviewer's Responses to Questions

**Comments to the Author**

1. Is the manuscript technically sound, and do the data support the conclusions?

Reviewer #1: Yes

Reviewer #2: Partly

2. Has the statistical analysis been performed appropriately and rigorously? 

Reviewer #1: Yes

Reviewer #2: No

3. Have the authors made all data underlying the findings in their manuscript fully available?

Reviewer #1: Yes

Reviewer #2: No

4. Is the manuscript presented in an intelligible fashion and written in standard English?

Reviewer #1: Yes

Reviewer #2: No

5. Review Comments to the Author

Reviewer #1: 1. Discuss the main motivations of the current work.

2. List out the man contributions of the current work.

3. Summarize the drawbacks of existing works in the form of a table.4. Some of the recent works such as the following can be discussed in the paper: "Image-Based malware classification using ensemble of CNN architectures (IMCEC), Deep learning and medical image processing for coronavirus (COVID-19) pandemic: A survey, Hand gesture classification using a novel CNN-crow search algorithm".

4. Compare the current work with recent state-of-the-art.

5. Discuss the limitations of the current work.

Reviewer #2: Please Check the format of the paper,

Update the paper lot of Typo's in the whole paper.

Please cite:

1. Krishna Kagita, M. and M. Varalakshmi, 2020. A detailed study of security and privacy of Internet of Things (IoT). International Journal of Computer Science and Network, 9 (3): 109–113.

2. Krishna Kagita, M. (2019). Security and Privacy Issues for Business Intelligence in a loT. In Proceedings of 12th International Conference on Global Security, Safety and Sustainability, ICGS3 2019 [8688023]

3. Navod Neranjan Thilakarathne, Mohan Krishna Kagita, Dr. Thippa Reddy Gadekallu. (2020). The Role of the Internet of Things in Health Care: A Systematic and Comprehensive Study. International Journal of Engineering and Management Research, 10(4), 145-159.

Visit this profile and cite related papers: https://scholar.google.com/citations?user=dz4QKZIAAAAJ&hl=en

6. PLOS authors have the option to publish the peer review history of their article (what does this mean?). If published, this will include your full peer review and any attached files.

Reviewer #1: No

Reviewer #2: No

---

## [Author Response · Author response to Decision Letter 0]

26 May 2021

EDITOR SUGGESTIONS:

Please ensure that your manuscript meets PLOS ONE's style requirements, including those for file naming.

RESPONSE: Thank you for your comments. We have addressed this issue by modifying the manuscript according to PLOS ONE’s style.

1. [First reviewer comment]

Discuss the main motivations of the current work.

RESPONSE: Thank you for your comment. We have discussed our motivation in the introduction: The goal of this study is to see if CNN can accurately discriminate medical images from their imaging modalities and anatomical location after they have been converted into other image format, namely JPEG format.

List out the main contributions of the current work.

RESPONSE: Thank you for your comment. We have discussed the main contribution of our work in the abstract section and in the conclusion section by saying: Our study has shown that using CNN for medical image classification is a promising methodology and could work on non-DICOM images, which could potentially save image processing time and storage space. 

Also, in our discussion section, we have stated that “despite all previous works, to our knowledge, there were no previous studies that used CNN to classify medical images by both image modality and anatomical location, which is most practical in both clinical and research setting”, which is another main contribution of our work.

Summarize the drawbacks of existing works in the form of a table.4. Some of the recent works such as the following can be discussed in the paper: "Image-Based malware classification using ensemble of CNN architectures (IMCEC), Deep learning and medical image processing for coronavirus (COVID-19) pandemic: A survey, Hand gesture classification using a novel CNN-crow search algorithm".

RESPONSE: This is an important comment and thank you for the reference you have provided. We had a cited this article “Deep learning and medical image processing for coronavirus (COVID-19) pandemic: A survey” in our work. We have also made a new summary table (Table 7) of the references mentioned in our discussion section that are related to medical image classification as you suggested. In our discussion section, we have also discussed regarding the drawbacks of these previous work that we have mentioned. 

Compare the current work with recent state-of-the-art.

RESPONSE: Thank you for your comment. Our discussion section had compared the current work with the previous studies that focused on medical image classification on either anatomy and/or image modality. We have also made a summary table (Table 7) of these references in this revision. 

Discuss the limitations of the current work.

RESPONSE: Thank you for your comment. We have mentioned several limitations regarding our work in the discussion section. Firstly, we only trained and tested on JPEG images for our CNN architecture. Secondly, we did not perform any image pre-processing for this study and only used the preset window level for our model. Thirdly, we did not choose all the sequences in the two MRI classes. Fourthly, a limited number of test set examples were included in this study. Fifthly, only 4 categories out of two image modalities and three anatomical locations were selected in this study. Please see the discussion section for a more detailed description of our work’s limitation.

2. [Second reviewer comment]

Please Check the format of the paper

RESPONSE: Thank you for this suggestion. We have modified the format of this paper according to PLOS ONE’s style.

Update the paper lot of Typo's in the whole paper.

RESPONSE: Thank you for this suggestion. We have revised our work and corrected the typos that we have found during this proofread. 

Please cite:

1. Krishna Kagita, M. and M. Varalakshmi, 2020. A detailed study of security and privacy of Internet of Things (IoT). International Journal of Computer Science and Network, 9 (3): 109–113.

2. Krishna Kagita, M. (2019). Security and Privacy Issues for Business Intelligence in a loT. In Proceedings of 12th International Conference on Global Security, Safety and Sustainability, ICGS3 2019 [8688023]

3. Navod Neranjan Thilakarathne, Mohan Krishna Kagita, Dr. Thippa Reddy Gadekallu. (2020). The Role of the Internet of Things in Health Care: A Systematic and Comprehensive Study. International Journal of Engineering and Management Research, 10(4), 145-159.

Visit this profile and cite related papers: https://scholar.google.com/citations?user=dz4QKZIAAAAJ&hl=en

RESPONSE: Thank you for your suggestion. We have read all three papers that you suggested. However, as these 3 papers were mainly regarding Internet of Things (IoT), which had little relevance to our work, which is about using CNN for medical image classification on both image modality and anatomical location, therefore, we did not cite these papers in this revised version. 

CONCLUDING REMARKS: 

Again, thank you for giving us the opportunity to strengthen our manuscript with your valuable comments and queries. We have worked hard to incorporate your feedback and hope that these revisions persuade you to accept our submission.

---

## [Decision Letter · Decision Letter 1]

31 May 2021

Automatic Classification of Medical Image Modality and Anatomical Location Using Convolutional Neural Network

PONE-D-21-11867R1

Dear Dr. Chiu,

We’re pleased to inform you that your manuscript has been judged scientifically suitable for publication and will be formally accepted for publication once it meets all outstanding technical requirements.

Kind regards,

Thippa Reddy Gadekallu

Academic Editor

PLOS ONE

Additional Editor Comments (optional):

Reviewers' comments:

Reviewer's Responses to Questions

**Comments to the Author**

1. If the authors have adequately addressed your comments raised in a previous round of review and you feel that this manuscript is now acceptable for publication, you may indicate that here to bypass the “Comments to the Author” section, enter your conflict of interest statement in the “Confidential to Editor” section, and submit your "Accept" recommendation.

Reviewer #1: All comments have been addressed

Reviewer #2: All comments have been addressed

2. Is the manuscript technically sound, and do the data support the conclusions?

Reviewer #1: Yes

Reviewer #2: Yes

3. Has the statistical analysis been performed appropriately and rigorously? 

Reviewer #1: Yes

Reviewer #2: Yes

4. Have the authors made all data underlying the findings in their manuscript fully available?

Reviewer #1: Yes

Reviewer #2: Yes

5. Is the manuscript presented in an intelligible fashion and written in standard English?

Reviewer #1: Yes

Reviewer #2: Yes

6. Review Comments to the Author

Reviewer #1: The authors have addressed all the suggestions and comments I recommend accepting the paper in its present form.

Reviewer #2: The authors have addressed all the problems proposed in the previous review round.

7. PLOS authors have the option to publish the peer review history of their article (what does this mean?). If published, this will include your full peer review and any attached files.

Reviewer #1: No

Reviewer #2: No

---

## [Editor Report · Acceptance letter]

3 Jun 2021

PONE-D-21-11867R1 

Automatic classification of medical image modality and anatomical location using convolutional neural network 

Dear Dr. Chiu:

I'm pleased to inform you that your manuscript has been deemed suitable for publication in PLOS ONE. Congratulations! Your manuscript is now with our production department. 

Kind regards, 

on behalf of

Dr. Thippa Reddy Gadekallu 

Academic Editor

PLOS ONE